# Recent Advances in the Synthesis, Characterization, and Application of Carbon Nanomaterials for the Removal of Endocrine-Disrupting Chemicals: A Review

**DOI:** 10.3390/ijms232113148

**Published:** 2022-10-29

**Authors:** Ze Liao, Yang Zi, Chunyan Zhou, Wenqian Zeng, Wenwen Luo, Hui Zeng, Muqing Xia, Zhoufei Luo

**Affiliations:** 1College of Bioscience and Biotechnology, Hunan Agricultural University, Changsha 410128, China; 2Hunan Provincial Key Laboratory of Phytohormones and Growth Development, Hunan Agricultural University, Changsha 410128, China

**Keywords:** endocrine-disrupting chemicals, carbon nanomaterials, synthesis, adsorption, application

## Abstract

The large-scale production and frequent use of endocrine-disrupting chemicals (EDCs) have led to the continuous release and wide distribution of these pollutions in the natural environment. At low levels, EDC exposure may cause metabolic disorders, sexual development, and reproductive disorders in aquatic animals and humans. Adsorption treatment, particularly using nanocomposites, may represent a promising and sustainable method for EDC removal from wastewater. EDCs could be effectively removed from wastewater using various carbon-based nanomaterials, such as carbon nanofiber, carbon nanotubes, graphene, magnetic carbon nanomaterials, carbon membranes, carbon dots, carbon sponges, etc. Important applications of carbon nanocomposites for the removal of different kinds of EDCs and the theory of adsorption are discussed, as well as recent advances in carbon nanocomposite synthesis technology and characterization technology. Furthermore, the factors affecting the use of carbon nanocomposites and comparisons with other adsorbents for EDC removal are reviewed. This review is significant because it helps to promote the development of nanocomposites for the decontamination of wastewater.

## 1. Introduction

### 1.1. Endocrine-Disrupting Chemicals and Their Damaging Effects

With the rapid development of the global economy, the release of endocrine-disrupting chemicals (EDCs) into the environment has become continuous. The International Security Program (IPCS) of the World Health Organization (WHO) has defined EDCs as exogenous substances or mixtures that alter the function(s) of the endocrine system and consequently adversely affect the health of populations or subpopulations of intact organisms or their progeny [1,2]. EDCs bind to receptors by imitating natural hormones and thereby interfere with the metabolism of the organism. There are two categories of EDCs: natural and synthetic. Natural EDCs are found in animals and humans and include genistein, coumarins, estradiol, and adrenocortical hormones [3,4]. Synthetic EDCs are classified into a variety of categories based on their application, chemical, and structural characteristics, such as pharmaceutical, organochlorine pesticides, detergents and surfactants, heavy metals, and dyes. The primary types and sources of synthetic EDCs are shown in Figure 1. Therefore, regardless of their class, substances exhibiting estrogenic activity in organisms can be considered as EDCs.

The public concern about the impacts of EDCs on both human health and the environment has grown particularly, owning to their wide occurrence, high persistence, bioaccumulation capabilities, and high toxicity. The main sources of synthetic EDCs are derived from the anthropogenic activity and industrial consumption of insecticides, plastic products, food packaging materials, electronics and construction materials, personal care items, medical tubing, antibacterial agents, detergents, cosmetics, pesticides, fabrics, and clothes [5]. In polar regions, EDCs were deposited by the “global distillation effect” and “the grasshopper effect” due to their semi-volatile nature and degradation resistance, leading to a global spread of contamination. The concurrence of EDCs was found in the Arctic and Antarctic regions and alpine regions [6,7]. The altitude dependence of polychlorinated biphenyls (PCBs) and polybrominated diphenyl ethers (PBDEs) was increased along a high-altitude aquatic food chain in the Tibetan Plateau, China [8]. The bioaccumulation of EDCs in aquatic organisms, especially fish, mussels, and daphnia, is an important criterion in risk assessment. The majority of organochlorine pesticides (OCPs) including DDTs could be enriched from the surrounding media into aquatic organisms through bioamplification of the food chain [9]. EDCs and their intermediate compounds are toxic and have harmful effects on the environment and human health. After an organism is exposed to estrogen, unstable intermediates with strong reactivity are generated by the biological metabolic enzymes and signalling transfer process [10]. Some of the metal ions of EDCs are covalently combined with cellular polymer components such as proteins and nucleic acids, resulting in irreversible chemical modifications [11]. Bisphenol A (BPA) is a ubiquitous EDC that has recently been associated with adverse effects on human health, precocious puberty, and sexual dysfunction [12,13]. Examples of synthetic EDCs and their negative impacts were show in Table 1. These organic molecules will enter the water cycle through rivers, lakes, or groundwater infiltration; the river’s organic chemicals flow into the ocean, while these pollutant EDCs enter the ocean via seawater intrusion. These EDCs are widely considered to be the major alarming sources of EDCs of great concern for the aquatic ecosystem.

EDCs are discharged into aquatic ecosystems via surface runoff, natural circulation, and the food chain. In particular, EDCs accumulate as they travel up the food chain, finally becoming harmful to human health. Low levels of EDCs cause neuroendocrine disorders, developmental malformations, and reproductive disorders [51,52]. Organisms such as aquatic plants, aquatic animals, and humans are at risk of long-term exposure to EDCs in the ecological environment. The accumulation of bisphenol A inhibits the growth of aquatic rice seedlings [53]. After exposure to some steroid EDCs, the expression of the VTG-A gene was disrupted in freshwater fish, leading to reproductive and developmental disfunctions, as well as decreased hatchability and reduced vitellogenin levels [54]. EDCs may lead to the feminization of some aquatic organisms, as well as metabolic diseases, cardiovascular risks, and reproductive issues in humans [55].

### 1.2. Distribution and Regularity of EDC Pollution

With the worldwide growth of manufacturing industries, EDC pollution in aquatic ecosystems is becoming more serious. In a 2008–2010 study of 46 representative active EDCs in 182 rivers in the United States by the U.S. Environmental Protection Agency, the approximate concentrations of steroid EDCs in surface waters reached 63 ng/L [56]. Aquatic plants such as algae absorb alkylphenols from polluted soil or water through their roots; EDC accumulation in algae was as high as 53 μg/L in the Bohai Sea region of China [57]. Eleven phthalate esters (PAEs) were measured in urine and serum samples collected from males diagnosed with infertility in Tianjin, China, and the median levels of mPAEs in the serum (n.d. to 3.63 ng/mL) were 1–2 orders of magnitude lower than the median levels of mPAEs in the urine (n.d. to 192 ng/mL) [58]. The bioaccumulation of EDCs harms the growth and development of animals and plants and endangers human health.

### 1.3. Treatment of EDC Pollution

Owing to the strong degradation resistance and high toxicity of EDCs, it is challenging to remove EDCs from wastewater. EDC removal methods mainly include chemical/electrochemical treatments, biological treatments, and physical treatments. Chemical/electrochemical methods mainly include chlorination, ozonation, photocatalysis, electrocoagulation, and anodic oxidation. Chemical/electrochemical treatments do not reduce the generation of chlorine residues or several other harmful intermediate products [59]. Biological treatments are widely applied in the secondary processes of sewage treatment plants and include aerobic and anaerobic treatments. However, these methods may produce degradation pollutants which increase the ecological risks of sewage treatments. Gadd et al. [60] reported that anaerobic treatments increased estrogenic content and estrogenic activity in cowshed wastewater. Compared with the former two technologies, physical adsorption is considered a more environmentally friendly wastewater treatment. The physical adsorption method is effective, efficient, and economical, and different adsorbents can be designed for use in this treatment method. Nanocomposites may represent a useful new wastewater treatment method; this treatment is advantageous due to the large specific surface area and high adsorption capacity.

## 2. Development of Carbon Nanomaterials

In recent years, carbon has become one of the most multilateral elements present in the periodic table due to its strength and ability to form bonds with other elements. Carbon-based nanomaterials and associated modified composites have excellent adsorption properties due to their environmentally friendly, extremely high specific surface area, large pore volume, uniform microporosity, and adjustable surface chemical properties [61,62]. Biomass carbon nanocomposites are a low-cost, readily available, widely distributed, and renewable nanomaterial. Several promising reports have emerged in recent years on the synthesis of carbon nanomaterials from cost-effective, rich, and renewable biomaterial resources such as saw dust, crab shells, bagasse, olive stone waste, and activated carbon cloth [63,64]. Activated carbon is widely used for the control of synthetic and naturally occurring EDCs in drinking water [65]. Carbon nanotubes (CNTs), which were discovered back in the year 1991, have been extensively adapted to study the adsorption capability in water treatment [66]. Graphene, a new material for which the Noble Prize was won, has received increasing attention due to its unique physico-chemical properties for removing EDC pollutants from wastewater [67]. Carbon dots with abundant functional groups (-OH, -COOH, -C=O) on their surface were specially designed to enhance the adsorption capacity [68]. Highly porous carbon sponges always contain some functional groups, which could enhance the surface sensitivity and selectivity of EDC pollutants [69]. Many carbon nanocomposites have been synthesized and used as adsorbents for the removal of EDC pollutants from wastewater. There are many industries such as mining, battery manufacturers, the pharmaceutical industry, the cultivation industry, galvanization, and metal finishing which generate wastewater containing EDCs and emit it directly or indirectly into the nearest water resources [5]. Carbon nanomaterials play an important role in nanoadsorbents. A schematic diagram of carbon nanoadsorbents which are used in EDC pollution treatments is illustrated in Figure 2. Carbon nanocomposites can be classified into seven categories including carbon nanofibers, carbon nanotubes, graphene family, magnetic carbon nanomaterials, carbon nanomembranes, carbon dots, and carbon sponges depending on their composition, structure, and characteristics. The seven categories of carbon nanocomposites according to their specific characteristics are depicted in Table 2.

### 2.1. Carbon Nanofiber

Carbon nanofiber (CN), also known as nano-activated carbon, is widely used for the treatment of organic wastewater and removal of EDC substances. In addition to the high adsorption capability, the regeneration of ACF could be carried out under a lower temperature than granular activated carbon [90]. The adsorption efficiency of ACF is significantly higher than that of granular or powdered activated carbon. Murayama et al. used ACF to recover organic chlorine pesticides (OCPs) from rainwater, river water, and seawater samples and confirmed that ACF adsorbed sub-ng/L level OCPs from environmental water samples [70].

### 2.2. Carbon Nanotubes (CNTs)

CNTs are a kind of carbon allotrope with an aromatic surface rolled up to form a cylindrical structure; the length of CNTs varies from 10 s of nm to 10 s of mm [76]. They can be divided into single-walled carbon nanotubes (SWCNTs) and multiwalled carbon nanotubes (MWCNTs), both of which have high a mechanical strength and elasticity as well as chemical and structural stability. SWCNTs and MWCNTs are widely used in removing EDCs from wastewater after being modified by different functional groups (polydopamine, magnetic particles, and molecular imprinting polymers e.g.,) [91]. Mashkoor et al. [92] summarized the adsorption capacities of the original and modified CNTs for different dyes; the adsorption capacity of glycine-*β*-cyclodextrin MWCNTs for methylene blue was up to 90.90 mg/g, while the adsorption capacities of modified MWCNTs for Hg^2+^, Cr^4+^, and As^3+^ were 123.45 mg/g, 146.5 mg/g, and 133.33 mg/g, respectively [74]. The large specific surface area, large pore volume, and average pore diameter contribute to the excellent adsorption capacity of CNTs [93].

### 2.3. Graphene Family

Graphene and graphene oxide (GO) have also been widely used for the adsorption of EDCs because of their special properties such as extraordinary quantum hall effects, high mobility, excellent electronic and mechanical properties, specific magnetism, and high thermal conductivity [94]. The excellent adsorption capacity of GO is improved by the electrostatic interaction between the oxidation groups and the adsorbates, and GO is mainly used for the adsorption of metal ions, anionic dyes, and cationic dyes. Modified GO has a good adsorption capacity for different heavy metals (60.2–1076.65 mg/g) [78]. GO also has a specific adsorption capacity for steroids: the removal efficiency of 17α-ethinylestradiol by magnetic GO can reach more than 25 mg/g [95].

### 2.4. Magnetic Carbon Nanocomposites

Various magnetic nanocomposites, including nanomaterials such as magnetic carbon nanotubes, magnetic graphene oxide, and magnetic metal–organic frameworks, exhibit excellent adsorption capabilities [96]. The advantage of magnetic nanomaterials is that their magnetic properties allow them to be separated from solutions and regenerated using an external magnetic field. Recently, magnetic nanocomposites have been extensively applied to the adsorption of EDC pollutants such as metals, steroid hormones, alkylphenols, and dyestuff from water [80,81]. Nanocomposites are primarily magnetized using ferriferous oxide (Fe_3_O_4_) and manganese ferrite (MnFe_2_O_4_). Different modified magnetic nanomaterials have different adsorption properties for specific anions, cations, and macromolecular dyes [97]. After the modification of PS-EDTA resin with magnetic ferric oxide nanoparticles, the Cr adsorption rate reached 99.3% [98]. An amino-functionalized mesoporous silica-magnetic GO nanocomposite, which was synthesized using a unique magnetic nanomaterial, removed oxytetracycline more effectively than the original magnetic graphene oxide (mGO) [99].

### 2.5. Carbon Membranes

Carbon membranes have become one of the most important materials employed in wastewater treatment. Carbon membranes have a low production cost and are highly permeable, selective, and stable [100]. Carbon membrane techniques are used for adsorption and filtration in a variety of wastewater treatment plants. The retention rates of nanoparticle-modified polyamide membranes for bisphenol A reached 99.4% [101]. The adsorption capacity of biopolymer carbon membranes modified with lignin, oat, soybean protein, sodium alginate, and chitosan for Pb^2+^ was 35 mg/g [102]. In addition, the adsorption capacity of an electrospun lignin–carbon membrane for methyl blue was 10 times greater than that of active carbon [103].

### 2.6. Carbon Dots

Carbon dots are single-layer or multilayer graphite structures or polymer aggregate carbon particles with a diameter less than 10 nm. Carbon dots have been acknowledged as discrete, quasispherical, fluorescent carbon particles, most of which are sp^2^ or sp^3^ hybrid carbon structures [104]. According to their structure and composition characteristics, carbon dots could be divided into graphene quantum dots (GQDs), carbon quantum dots (CQDs), and carbonized polymer dots (CPDs). The synthesized carbon dots show water solubility, good chemical stability, low toxicity, and good biocompatibility. In the adsorption treatment of water pollution, GQDs have received increasing attention as excellent adsorbents. It was reported that GQDs have good adsorption properties in removing pesticides, dyes, heavy metals, and drugs from water [68,105]. Modifying GQDs improves the adsorption effect of nanocomposites by increasing the specific surface area for removing anionic and cationic dyes. Meanwhile, the oxygen-containing functional groups on GQDs also enhance the electrostatic interaction between dyes and nanocomposites. It is the contribution of the formation of hydrogen bonds to the surface between the adsorbate and GQDs. The removal efficiency of Hg^2+^ and Pb^2+^ by GQD adsorption were 98.6 and 99.7%, respectively [86]. The other kinds of carbon dot nanocomposites also have an excellent EDC removal ability. For example, the maximum adsorption capacities of carbon dot nanocomposites for carbamazepine and tetracycline could reach up to 65 mg/g and 591.72 mg/g, respectively [87]. GQDs present significant opportunities for the adsorption of EDCs and pose challenges for future work in environmental application fields.

### 2.7. Carbon Sponges

Carbon sponges are a spongy nanomaterial with a high temperature resistance, excellent elasticity, fast adsorption rate, and low cost [106,107]. Graphene-based and carbon nanotube-based materials with aerogel structures were developed in recent years for various adsorption applications. Graphene aerogels and carbon nanotube aerogels are macroscopic porous materials with a unique isotropic structure. Dubey et al. [108] synthesized the graphene aerogel, which is the lightest material with a density of 0.16 g/L, in 2013. Graphene aerogels have a high adsorption efficiency for EDC pollutants such as dyes. For example, Li et al. [89] reported a new graphene sponge by using polyvinyl alcohol cross-linked GO for adsorbing methylene blue. The results showed that the excellent adsorption performance for the GO/polyvinyl alcohol sponge captured methylene blue in the flow state. In addition, carbon nanotube sponges are widely used to treat water pollution. Wang et al. [106] used carbon nanotube sponges as adsorbents to enrich trace polychlorinated biphenyls (PCBs) in water samples; the recovery rates of this analysis method for carbon nanotube sponges for PCBs are 81.1% to 119.1%, which have a good application prospect.

## 3. Innovative Methods of Carbon Nanocomposite Synthesis

Carbon nanocomposite synthesis methods can be divided into four categories according to the reaction conditions and formation processes used: in situ polymerization, direct compounding, solvothermal synthesis, and electrospinning.

### 3.1. In Situ Polymerization

In situ polymerization is the most common method of nanocomposite synthesis. In situ polymerization prevents the agglomeration of nanocomposites and results in uniform distributions and structures [109]. Youssef et al. [110] added prepared polyvinyl alcohol (PVA) to a solution of MWCNT materials, stirring for 3 h at room temperature, and dripping in 0.01 g of citric acid over 30 min to obtain an MWCNT/PVA hybrid polymer nanocomposite film. The addition of 6% MWCNT to PVA resulted in a remarkable increase in tensile strength (to 11.45 MPa) prior to the formation of the nanocomposite film. The removal rates of Cr^2+^, Cd^2+^, and Pb^2+^ by this composite material were more than 90%. In addition, the adsorption performance of carbon nanomaterials was improved after modification using in situ polymerization, and this method may thus play an important role in environmental applications.

### 3.2. Direct Compounding

Direct compounding is a synthesis method that does not require an activator or special reaction conditions. Direct compounding is widely used because of its simple steps, relatively low cost, and ease of expansion. For example, nanocomposites can be synthesized directly by mixing the polymer matrix and the nanomaterials under mechanical forces [80]. Yang et al. [111] synthesized an anionic polyacrylamide-functionalized GO composite by pouring an anionic polyacrylamide solution into a GO solution with slight mechanical stirring. The maximum adsorption capacity of this composite for basic fuchsin was up to 1034.3 mg/g, which indicates that anionic polyacrylamide/GO aerogels are promising adsorbents for the removal of dye pollutants from aqueous solutions.

### 3.3. Solvothermal Synthesis

Solvent thermal synthesis, for which water or organic solvents are the reaction medium, is usually used in a closed environment to create a critical reaction state (i.e., high temperature and high pressure) to generate homogeneous nanomaterials. Because water molecules hydrolyze at high temperatures, solvothermal synthesis has the advantage of forming nanoparticles of ideal shape, uniform size, and specific surface area. In solvent thermosynthesis, a molecule can easily be functionalized by simply replacing the original organic ligand with functional groups. Solvothermal synthesis can be finished in one step without adding a catalyst, such as the green synthesis of functional GO. Ammonia-modified GO was synthesized by solvothermal reaction at 180 °C for 10 h [112]. Guo et al. [113] synthesized Fe_3_O_4_–GS composites by solvothermal reaction by mixing FeCl_3_·6H_2_O, ethylenediamine, and GO at 200 °C for 8 h.

### 3.4. Electrospinning

Electrospinning is a useful fabrication technique that produces NF membranes. Electrospinning relies on the electrostatic repulsion between surface charges to absorb nanofibers in a viscoelastic fluid. For example, Sun et al. [114] prepared a super-hydrophobic carbon fiber membrane by electrospinning. Carbon nanofibers were prepared by electrospinning for their special phase morphology, crystal structure, and surface geometry. Electrospinning provides a simple and versatile method for generating ultrathin fibers from a rich variety of materials that include polymers, composites, and ceramics. Electrospinning synthesis technology can greatly improve the production efficiency of carbonyl nanomaterials.

## 4. Characterization Techniques

### 4.1. Morphological and Microstructural Analysis

Adsorption rates are related to nanocomposite porosity, particle size, and other factors. The surface morphology of a nanocomposite can be observed using scattering electron microscopy (SEM), transmission electron microscopy (TEM), and atomic force microscopy (AFM) [115,116]. SEM and AFM more accurately reflect grain size and other surface properties as compared to other available techniques, but TEM is commonly preferred due to its high resolution. TEM provides two-dimensional (2D) projected images of three-dimensional (3D) objects, directly revealing the size and shape of nanocomposites. Zhang et al. [117] analysed the specific shape and micromorphology of carbon/SiO_2_ core-sheath nanofibers by combining TEM and SEM technologies. The physical microstructure and compositional elements of the nanomaterial surface can be observed using various techniques, including SEM-energy dispersive X-ray spectroscopy (EDX), TEM-EDX, SEM-elemental mapping, ultraviolet–visible spectroscopy (UV-VIS), X-ray diffraction (XRD), Fourier transform infrared (FT-IR), and X-ray photoelectron spectroscopy (XPS). Wang et al. [118] investigated the surface morphologies and chemical compositions of carbon nanofibrous membranes using SEM-EDX. EDX detectors are equipped with ultra-thin element light windows that detect elements. XRD can be used to identify the structures of diffraction patterns based on various indexes such as element type. The functional groups of nanocomposites were identified using FT-IR spectroscopy. Wu et al. [119] analyzed the formation of Ag nanoparticle decorating in carbon nanotube sponges by FT-IR characterization. Surface area and morphology are important characteristics that affect the adsorption of organic pollutants. Raman spectroscopy is a vibrational spectroscopic technique that can provide detailed information about the structure, chemical composition, and morphology of nanocomposites. Raman spectroscopy is an important method of GO composite characterization. The methods used to synthesize and characterize some carbon nanocomposites are presented in Table 3.

### 4.2. Magnetic Properties

Magnetism is also useful for adsorbent separation. Adsorbents can be easily separated from solutions without secondary contamination using an external magnetic field. Superconducting quantum interference devices (SQUIDs), alternating gradient magnetometers (AGMs), and vibrating sample magnetometers (VSMs) are commonly used to measure the magnetization and coercivity of magnetic nanocomposites [82]. SQUID measurements were used to investigate the magnetic properties of pure and Mn-doped electrospun barium titanate nanofibers. VSM plays a very important role in the measurement of the magnetization of magnetic nanocomposites. VSM measurements were used to investigate magnetic nanofibers containing ferromagnetic Ni nanoparticles. The significant hysteresis loops in the S-shaped curves indicate the ferromagnetic behaviors of the nanocomposites. In addition, VSM differentiates between measurements parallel and perpendicular to the axes of the aligned nanofibers. VSM has also been used to measure the effects of magnetic material doping on magnetization, the squareness of hysteresis loops, and the straightening force of materials such as samarium–cobalt nanofibers [80].

## 5. Application in Wastewater

Several recent studies have investigated the real-world application of nanoadsorbent materials in wastewater treatment. Material regeneration and reuse in real-word applications and comparisons of nanoadsorbents with other common adsorbent materials are summarized below.

### 5.1. Regeneration and Reuse

Due to the economic benefits of EDC pollution treatments, it is necessary to summarize the regeneration and reuse efficiency of various nanoadsorbents.

A variety of adsorbent regeneration technologies have been developed, including thermal, steam, chemical, microwave-assisted, electrochemical, and biological regeneration methods. It was shown that thermally regenerated MWCNT (incubated at 300 °C for 2 h) maintained its adsorption efficiency for cyclophosphamide, ifosfamide, and 5-fluorouracil in water. These results indicate that, when regenerating an adsorbent, a suitable regeneration technique should be selected according to the chemical and physical properties of the adsorbent itself [123]. The adsorption rates of neutral and Pb^2+^ by the MnFe_2_O_4_/GO nanocomposite were 94% and 98.8%, respectively, and these magnetic nanomaterials could be recycled five times with good stability [124].

Generally, the adsorption efficiency of a recycled carbon nanoadsorbent will decrease as the number of use-cycles increases. Carbon-based nanocomposites have a better regeneration ability. In studies on the adsorption removal of rhodamine B dye using GO-based nano-nickel composite materials, the adsorption efficiency decreased from 90% to 85% after the first recovery and decreased to 60% after the fifth recovery [125].

### 5.2. Carbon Nanoadsorbent Patents

Patented carbon nanocomposites were reflected upon to understand the development and applicability of carbon nanoadsorbents in wastewater treatment. Dai et al. prepared a novel GO/chitosan composite adsorbent (Dai et al., patent CN113318710A. 31 August 2021). The adsorption capacity of the GO/chitosan composite adsorbent for Cr^6+^ was significantly enhanced. Guo et al. reported a novel method to synthesize a polyamidine/carbon nanomaterial that could be used to remove some anionic dyes in wastewater (Guo et al., patent CN109174035B. 15 June 2021). A novel modified carbon nanotube was prepared by Huang et al. for the adsorption of Pb^2+^. The sulfhydryl functional groups on the surface of carbon nanotubes could strongly chelate with Pb^2+^. The adsorption rate of Pb^2+^ can reach 92.89% (Huang et al., patent CN110449132B. 25 March 2022). Carbon sponge blending by nanocellulose, polyvinyl alcohol, and polyvinylpyrrolidone could efficiently remove Cr^6+^ (9.3948 mg/g) and organic dyes from water (Ma et al., patent CN105597681B. 14 November 2017). The adsorbing capacity of a nitrogen-doped graphene quantum dot hybrid membrane for Pb^2+^ removal is 9 mg/g (Liu et al., patent CN112723346B. 24 June 2022).

### 5.3. Comparison with Other Adsorbents

In Table 4, the EDC adsorbent efficacy among carbon nanocomposites and other materials based on previous studies was compared. Carbon nanocomposites show great promise owing to their many advantages: First, compared with other adsorbents, carbon nanocomposites have a larger surface area and better adsorption capacity [126]. For example, the adsorption capacity of most activated carbons for methyl blue is 100–500 mg/g, while the nanoadsorbent capacity for MB is often higher than 500 mg/g [127]. ACF derived from japonica seed hair fibers has an adsorption capacity for MB of 943.372 mg/g [128]. Second, the application of nanocomposites can greatly reduce the cost of water treatment. Beck et al. [103] reported that the use of carbon nanofiber membranes can reduce energy consumption by 87% during wastewater treatment processes. Third, some biodegradable polymer/multiwalled carbon nanotubes can effectively reduce environmental pollution and have good adsorption properties for EDCs [129]. These carbon nanomaterials can be biodegraded after adsorption and the nanomaterials can be recycled.

## 6. Factors Affecting the Adsorption of Pollutants on Carbon Nanocomposites

### 6.1. Effects of Solution pH

The solution pH value is an important factor affecting carbon nanocomposite adsorption capacity. The relationships among solution pH, nanocomposite pHpzc, and EDC pollutant pKa should be comprehensively analyzed to determine the effects on adsorption efficiency [97]. When the pH of the solution is less than the pH_pzc_ of the nanocomposite, the surface of the adsorbent is positively charged due to the protonation of the surface functional groups. Conversely, when the surface of the adsorbent is negatively charged, the electrostatic interactions between the adsorbent and the EDC pollutants are altered, which then affects the adsorption capacity of the nanoadsorbent on the EDCs [140]. For example, the removal efficiencies for MB by functionalized MWCNTs at a pH of 2, 4, and 6 were 65%, 85%, and 95%, respectively [141]. Kumar et al. [142] showed that phenolic compounds usually exist in a neutral form when the solution pH is less than the pKa of phenolic EDCs; however, the functional groups of the phenolic compounds dissociate and generate phenoxide groups when the solution pH is greater than the pKa of the EDC pollutant; thus, the phenolic EDCs are repelled by each other and the negatively charged adsorbent, reducing the removal efficiency of the adsorbent. These results show that the removal efficiency of the carbon nanoadsorbent for phenolic EDCs is maximized in solutions with a pH less than the pKa (~6).

### 6.2. Effects of Adsorbent Dosage and the Initial Concentration of EDC Pollutants

Adsorbent dosage and initial EDC concentration are usually considered the most important factors influencing liquid-phase adsorption. Adsorbent dosage is correlated with the adsorption site and capacity. Baharum et al. [143] investigated the effects of adsorbent dosage on the removal of polycyclic aromatic hydrocarbons and found that the adsorption capacity of diazinon decreased with increasing adsorbent dosage. This may have resulted from the aggregation of the active site due to the violent collisions among the excess bioadsorbent particles.

The initial EDC concentration strongly influences the adsorption capacity of the nanocomposite. At low EDC concentrations, the number of EDC molecules initially available to the adsorption sites is low. At high EDC concentrations, the ratio is again low due to the lack of available sites. Thus, the removal rate is affected by the initial EDC concentration. Hameed et al. [144] reported that, within a certain range, increases in the initial pesticide concentration increased the adsorption rate of activated carbon.

### 6.3. Effects of Adsorption Equilibrium Duration

The duration of the adsorption equilibrium between the nanoadsorbent and the EDC adsorbate is a vital parameter for evaluating the adsorption efficiency and time cost. In the initial state, the removal efficiency of the adsorbent usually increases with contact time until the adsorption process reaches equilibrium. The length of the period before the adsorption equilibrium is reached varies by nanocomposite, which greatly impacts the cost of sewage treatment. For example, the equilibrium time for Pb^2+^ removal by porous carbon nanofibers is 60 min, and the removal rate is up to 80% [145]. The equilibrium time for the removal of tetracycline by a copper alginate-carbon nanotube membrane is 2000 min, and the equilibrium adsorption capacity is 120 mg/g [146].

### 6.4. Effects of Temperature

Temperature is an important factor affecting the performance of the nanoadsorbent. In general, when the adsorption process is endothermic, the adsorption capacity increases with temperature; when the process is exothermic, the adsorption capacity decreases as the temperature increases [144]. For example, Yadav et al. [147] adsorbed CAC-500 onto phenol EDCs and found that the adsorption capacity of phenol decreased from 1.65 mg/g to 1.54 mg/g as the temperature increased from 25 °C to 55 °C. Because this adsorption process is exothermic, increasing the temperature reduced the adsorption capacity of the adsorbent.

## 7. Conclusions and Perspectives

EDCs are increasingly threatening human and aquatic life; therefore, treatment of these pollutants is of utmost importance. Adsorption has a wide applicability for the removal of EDC pollutants from wastewater. Carbon-based nanomaterials have been extensively explored for EDC adsorption applications because they are eco-friendly, their good chemical stability, structural diversity, low density, and suitability for large-scale production. This review is a timely summary of recent advances in the treatment of various types of EDC-polluted wastewater via carbon nanocomposite adsorption. This review details the synthesis methods, characterization, and explores the latest developments in carbon nanocomposites as adsorbents for EDC wastewater treatment. The comparisons of maximum adsorption capacities and costs of carbon nanocomposites are reviewed in this work with various adsorbents previously studied. While considering the aforementioned requirements and constraints, the present review critically affirms the capabilities of carbon nanomaterials in adsorbing EDC contaminants from wastewater.

Outlooks and challenges have discussed to inspire more exciting developments in the application of carbon nanocomposites for the remediation of wastewater. The properties of carbon nanoadsorbents increase their potential applicability, and carbon nanoadsorbents may be more useful and beneficial in several fields than other adsorbents. However, it should be pointed out that carbon nanomaterials as adsorbents still face some limitations. The preparation cost of some carbon nanomaterials is relatively high, such as carbon nanotubes and graphene. On the other hand, there are some technical problems with the recyclability of some carbon nanomaterials for large-scale wastewater treatment [148,149]. It is still a challenge to develop new, safe, efficient, and lower-cost carbon nanocomposite adsorbents. Moreover, future studies should further explore novel aspects of carbon nanomaterials, such as chemical stabilization and surface adaptations, to improve their applicability to the treatment of water and wastewater. The challenges involved in the development of these novel nanoadsorbents for the decontamination of wastewaters have also been examined to help identify future directions for this emerging field to continue to grow.

## Figures and Tables

**Figure 1 ijms-23-13148-f001:**
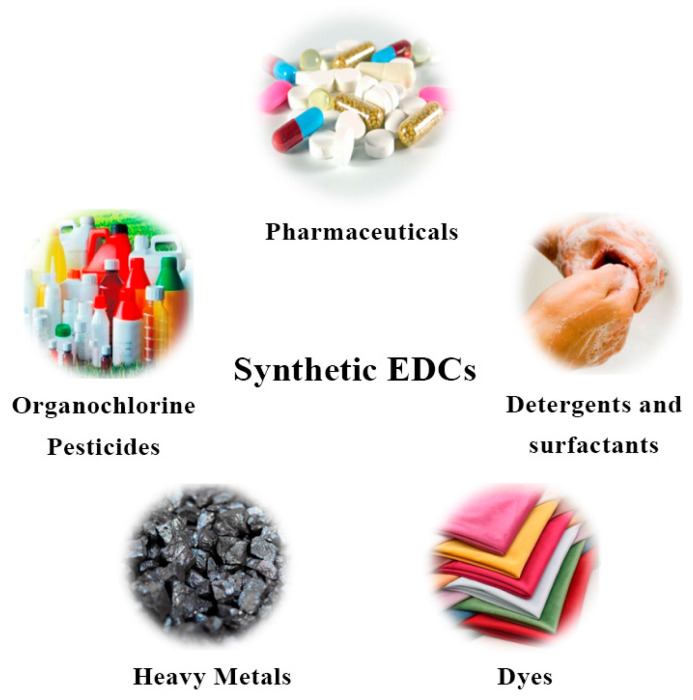
Different sources of synthetic EDCs.

**Figure 2 ijms-23-13148-f002:**
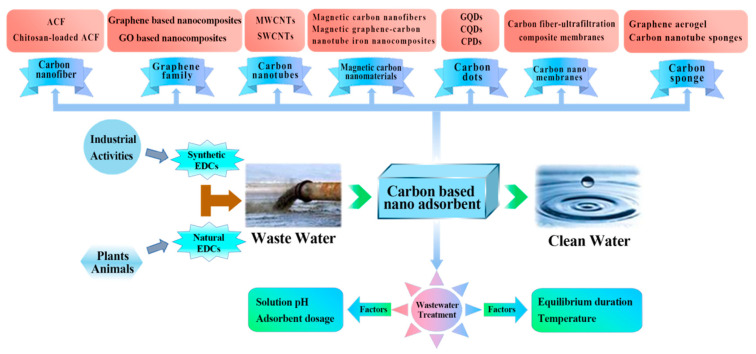
Schematic diagram of carbon nanoadsorbents for EDC adsorption for wastewater treatment, the categories of carbon nanoadsorbents, and nanoadsorbents in applications for the decontamination of wastewater.

**Table 1 ijms-23-13148-t001:** Examples of synthetic EDCs and their negative impacts.

EDC Class	Example	Negative Impact	Refs
Pesticides	Dichlorodiphenyltrichloro-ethane (DDT) *2*,*4*-dichlorophenoxyacetic acid(*2*,*4*-D) Polychlorinated biphenyl (PCBs) Methoxychlor (MXC) Cyhalothrin Organophosphate pesticide	Disturb hormonal balance in women, harm to the liver, Parkinson’s, preterm birth (PTB) metabolic syndrome, infertility, testosterone concentration reduction.	[14,15,16,17]
Steroidal pharmaceuticals	Estradiol Ethynylestradiol Testosterone Androstenedione Corticosteroids Progesterones	Osteonecrosis of the jaw, interferes with the thyroid endocrine system, placenta previa, lung cancer, stimulates lung adenocarcinoma cell production.	[18,19,20,21,22]
Detergents and Surfactants	Alkylphenols (APs) Bisphenol A Diethylstilbestrol (DES) Nonyphenol Alkylphenol Ethoxylates (APEO) Perfluorooctanoic acid (PFOA) Perfluorooctane sulphonate (PFOS) Alkylphenol carboxylates	Cardiovascular risk factors, ovarian, uterine, pituitary, testicular cancers, diabetes, low sperm count, recurrent miscarriages.	[23,24,25,26,27,28]
Personal care products	Benzophenone Oxybenzone Chlorophene *N*,*N*-Diethyl-m-toluamide (DEET) *tris*(*2*-chloroethyl)phosphate (TCEP)	DNA damage, obesity, hepatic steatosis.	[29,30,31]
Nonsteroid pharmaceuticals	Paracetamol Indomethacin Aspirin Ibuprofen Tetracycline	Wheeze and asthma risk in children, hepatocyte senescence, risk of severe intraventricular hemorrhage, interference with the endocrine activity of the thyroid gland.	[32,33,34,35,36]
Dyes	Methylene blue (MB) Aniline yellow Rhodamine B (RB) Thiazine	Reduce soil fertility and the photosynthetic activity of aquatic plants, potentially promoting toxicity, mutagenicity, and carcinogenicity.	[37,38,39]
Disinfection by-products	Haloacetamides Bromoacetonitriles Cyanoformaldehyde Bromoaldehydes	Induced liver and kidney injury, mammalian cell cytotoxicity and genotoxicity.	[40,41,42]
Heavy Metals	Ni^2+^, Pb^2+^, As^3+^, Cr^3+^, Hg^2+^, Cd^2+^	Prostate cancer, hepatotoxicity, nephrotoxicity, skeletal toxicity.	[43,44,45,46,47]
Industrial additives and agents	Polybrominateddiphenyl ethers (PBDEs) *2*,*3*,*7*,*8*-tetrachlorodibenzo- p-dioxin (TCDD) Polyfluoroalkyl substances (PFAS) Phthalate Esters (PAEs) Polyfluoroalkyl	Risk of diabetes, alteration of cognitive functions, risk of atherosclerosis, various cancers, elevated cholesterol levels, decreased immune and liver functionalities, birth defects.	[48,49,50]

**Table 2 ijms-23-13148-t002:** Different categories of carbon nanoadsorbents.

Classification	Category	Type of EDCs	Refs
Carbon nanofiber	Activated carbon fiber (ACF)	Organochlorine pesticides	[70]
	Chitosan-loaded activated carbon fiber	Pb^2+^-EDTA complex	[71]
	Activated carbon fiber supported/modified nanotubes	*2-*Chlorophenol	[72]
Carbon nanotubes	Multiwalled carbo nanotubes (MWCNTs)	Triton X-100, *17β-E*stradiol, Sodium dodecylbenzene, Hg^2+^, Cr^3+^, As^3+^, Bisphenol A	[73,74,75]
	Single-wall carbon nanotubes (SWCNTs)	Hg^2+^, Cr^3+^, As^3+^, Hexachlorocyclohexane, Dichloro-diphenyl-trichloroethane	[76]
Graphene family	Graphene oxide (GO) and other nanocomposites	Perfluoroalkyl substances Pb^2+^, Hg^2+^, As^5+^, Cr^6+^, Methylene blue	[77,78,79]
Magnetic carbon- nanomaterials	Magnetic carbon nanofibers	Phenol, Rhodamine B	[80]
	Magnetite/porous graphene-base nanocomposites	Cr^6+^, Pb^2+^, Cd^2+^, As^3+^, Polychlorinated biphenyl, Dye	[81]
	Magnetic graphene–carbon nanotubes iron nanocomposites	Pb^2+^, Cd^2+^	[82]
Carbon nanomembranes	Carbon fiber ultrafiltration composite membranes	Steroid hormones	[83]
Carbon dots	Graphene quantum dots (GQDs) Carbon quantum dots (CQDs) Carbonized polymer dots (CPDs)	carbamate pesticide oxamyl, Cd^2+^, Hg^2+^, Pb^2+^, tetracycline, Carbamazepine	[84,85,86,87,88]
Carbon sponge	Graphene sponges	Methylene blue, Pb^2+^	[89]
	Carbon nanotube sponges	Polychlorinated biphenyl	[69]

**Table 3 ijms-23-13148-t003:** Synthesis and characterization of some carbon nanoadsorbents.

Classification	Category	Characterizations	Synthesis	References
Carbon nano tubes (CNTs)	Activated carbon fiber-supported/modified titanate nanotubes	TEM, FE-SEM, XRD, FTIR, XPS, UV-vis	Hydrothermal method	[72]
Graphene	Graphene oxide (GO)	XPS, FTIR, AFM, SEM, TEM	Chemical reduction Hummer’s method	[77,78]
	Magnetic chitosan and graphene oxide (MCGO)	TGA, FTIR, TG, SEM, DSC	Hummer’s method Ultrasonic dispersion	[79]
Magnetic carbon-nanomaterials	Magnetic carbon nanotube iron nanocomposites	BET, XRD, XPS, TEM, SEM, AFM, SAED, μXRF	Solvothermal synthesis Microwave irradiation	[81]
	Magnetite/porous graphene-based nanocomposites	FE-SEM, EDX, XRD, FTIR, RAM, VSM, UV-vis, XPS, SSA	Sonication Solvothermal synthesis	[82]
	Electrospun magnetic carbon nanofibers	SQUID, AGM, VSM, MOKE	Electrospinning Field-assisted electrospinning Solvothermal technique	[80]
Carbon membranes	Carbon fiber ultrafiltration composite membranes	FE-SEM, BET, FTIR	Filtration process Sonication mixing	[83]
Carbon dots	Graphene quantum dots (GQDs) Carbon quantum dots (CQDs) Carbonized polymer dots (CPDs)	XRD, AFM, TEM, UV-vis, XPS, FTIR	Solvothermal technique Hydrothermal method	[120,121,122]
Carbon sponge	Graphene aerogel Carbon nanotube sponges	XRD, XPS, FTIR, FE-SEM	Solvothermal technique	[119]

Abbreviations: FE-SEM, field emission scanning electron microscope; XRD, X-ray diffractometer; TEM, transmission electron microscopy; FTTR, Fourier transform infrared; XPS, X-ray photoemission spectroscopy; UV-vis, ultraviolet–visible spectroscopy; AFM, atomic force microscopy; SEM, scanning electron microscopy; DSC, differential scanning calorimetry; VSM, vibrating sample magnetometer; BET, Brunauer–Emmett–Teller; RAM, Raman spectra; SQUID, superconducting quantum interference device; AGM, alternate gradient magnetometer; MOKE, magneto-optical Kerr effect; μXRF, micro X-ray fluorescence; SSA, specific surface area analysis.

**Table 4 ijms-23-13148-t004:** Comparison with other materials of EDC adsorbents.

Adsorbents	Advantages	Disadvantages	Refs
Carbon nanomaterials	Large specific surface, surface multi-functionality, regenerative capabilities, eco–friendly, high adsorption capacity, biodegradability	High cost to realize large-scale production and application	[130,131,132]
Biomass	Abundant in nature, available in large quantities, inexpensive, have potential as complexing materials	At the laboratory stage	[133,134]
Natural zeolites	Easily available and relatively cheap	Low permeability, continuous need for amendments (pH adjustment)	[135,136]
Conventional ion exchange resins	Effectiveness, ease of operation, large available exchange capacities, small footprint, regenerative capabilities	Mostly operated only in low pH ranges	[137,138]
Biodegradable polymers	Biodegradability, high local availability, low cost, high surface area, high chemical stability, remarkable flexibility	Risk of contamination, possible release of methane, costly regeneration	[129,139]

## Data Availability

The data presented in this study are available on request from the corresponding author.

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
