# Peer review of "Recent Advances in the Synthesis, Characterization, and Application of Carbon Nanomaterials for the Removal of Endocrine-Disrupting Chemicals: A Review"

_ijms, 2022, doi:10.3390/ijms232113148_

Round 1

Reviewer 1 Report

The manuscript focuses on a hot and interesting field. However, a coherent picture and conclusions on the subjects are lacking. It contains many general statements of little scientific value; authors often describe only one or two examples - without a general conclusion. The manuscript does not give a consistent picture of the topic as is required from a Review paper - I recommend a thorough revision of the manuscript. In its current form, I do not recommend publishing it in a newspaper of this quality.

Author Response

Dear reviewer:

We would like to thank the editor for giving us a chance to revise our paper, and also thank the reviewers for giving us constructive suggestions and comments which will help us to improve the quality of our submission. Here, we would like to submit our revised manuscript entitled “Recent advances in the synthesis, characterization, and application of nano-composites to the removal of endocrine-disrupting chemicals: a review”. Our response was attached below and the amendments have been highlighted on the revised manuscript.

Thank you for your re-evaluation of our revised submission. We hope that the revision will be acceptable for publication.

The manuscript focuses on a hot and interesting field. However, a coherent picture and conclusions on the subjects are lacking. It contains many general statements of little scientific value; authors often describe only one or two examples - without a general conclusion. The manuscript does not give a consistent picture of the topic as is required from a Review paper - I recommend a thorough revision of the manuscript. In its current form, I do not recommend publishing it in a newspaper of this quality.

Author reply: Thank the reviewer for giving us constructive suggestions which will help us to improve the quality of our submission. We tried our best to revise this manuscript according to your suggestions. The details could be found in revision manuscript.

Reviewer 2 Report

The manuscript entitled “Recent advances in the synthesis, characterization, and application of nano-composites to the removal of endocrine-disrupting chemicals: a review” aims to review the nano-composites that have been successfully used to remove EDCs from wastewater and also recent advances in nano-composite synthesis and characterization. The topic is important, but the manuscript is prepared poorly. There are many material errors within the manuscript.

Here are my specific concerns:

Organochlorine pesticides are not the only class of pesticides with endocrine disruptive potential. How did the authors choose?

Dichlorvos is an organophosphate pesticide.

Line 14: not characteristics techniques, but characterization techniques.

The motivation for the selection of materials type and EDCs is not clear.

Too many materials are described in brief. It would be better to narrow the choice to, for example, carbon materials and describe them in more depth. Written like it is now, the manuscript has little scientific value.

Adsorption mechanisms are not needed here.

Overall, the manuscript is too wide and not informative. The concept has to be changed. In the present form, it is the manuscript about everything but basically nothing. It should be completely rewritten and resubmitted.

Author Response

Dear reviewer:

We would like to thank the editor for giving us a chance to revise our paper, and also thank the reviewers for giving us constructive suggestions and comments which will help us to improve the quality of our submission. Here, we would like to submit our revised manuscript entitled “Recent advances in the synthesis, characterization, and application of nano-composites to the removal of endocrine-disrupting chemicals: a review”. Our response was attached below and the amendments have been highlighted on the revised manuscript.

Thank you for your re-evaluation of our revised submission. We hope that the revision will be acceptable for publication.

The manuscript entitled “Recent advances in the synthesis, characterization, and application of nano-composites to the removal of endocrine-disrupting chemicals: a review”aims to review the nano-composites that have been successfully used to remove EDCs from wastewater and also recent advances in nano-composite synthesis and characterization. The topic is important, but the manuscript is prepared poorly. There are many material errors within the manuscript.

Author reply: Thank the reviewer for giving us constructive suggestions and corrections which will help us to improve the quality of our submission. We tried our best to revise this manuscript according to your suggestions.

Specific comments:

  1. Organochlorine pesticides are not the only class of pesticides with endocrine disruptive potential. How did the authors choose?

Author reply: Thank the reviewer for giving us valuable advice. Previous manuscript on the classification of EDCs in pesticides were confusion. Organochlorine pesticides just one of pesticides which interfere with the endocrine system. "Organochlorine pesticides (OCPs)" in Table1 was revised to "Pesticides". The details could be found in Table 1.

2.Dichlorvos is an organophosphate pesticide.

Author reply: "Dichlorvos" was revised to "Organophosphate pesticide".The details could be found in Table 1.

3.Line 14: not characteristics techniques, but characterization techniques.

Author reply: "characteristics techniques" was revised to "characterization techniques". The details could be found at line 20.

4.The motivation for the selection of materials type and EDCs is not clear.

Author reply: Thank you for your kind suggestion. We are very sorry about that the motivation for the selection of materials type and EDCs is not indicate enough. We have further revised the article and made a detailed introduction as suggested. The motivation for the selection of EDCs and materials type were in-depth discussed and the details could be found at line 41-46 and line 92-101.

  1. Too many materials are described in brief. It would be better to narrow the choice to, for example, carbon materials and describe them in more depth. Written like it is now, the manuscript has little scientific value.

Author reply: We appreciate that the reviewer gives us so many constructive suggestion and feel sorry about the shortcoming of the depiction of carbon materials. The details of revision could be found at lines 115 to 143.

6.Adsorption mechanisms are not needed here.

Author reply: Thanks to the reviewers for your comments, we have thought about it carefully and feel that this part is important. It is necessary to understand the adsorption mechanism of removing EDCs by nano composites in order to apply them to actual wastewater. On the one hand, the adsorption mechanism has already become a focus in recent years. Many studies on the adsorption mechanism of nano adsorbents were reported (Mashkoor et al, Environmental Chemistry Letters, 2020; Samaneh Saber-Samandari et al., Chemical Engineering Journal, 2016; Muhammad Usman et al, Science of the Total Environment, 2021). Finally, we hope this part could be retained.

7.Overall, the manuscript is too wide and not informative. The concept has to be changed. In the present form, it is the manuscript about everything but basically nothing. It should be completely rewritten and resubmitted.

Author reply: Thank you very much for the valuable comments. We have carefully revised the article according to your comments. There revision details were as follows:

(1)“Activated carbon fiber supported/modified nanotubes” in Table 2 was moved to the classification of "Activated carbon fiber(ACF)";

(2)“Bimetallic TiO/CNT/Pd-Cu” in Table 2 was deleted;

(3)The part 6.1 about adsorption mechanism was moved to the part 5. The details could be found at line 375-396;

(4)“Nanoparticles” in Table 6 was changed to “Nano materials”;

(5) The abstract was revised.

Reviewer 3 Report

This is a very interesting article that will be of interest to a wide range of readers. The authors have done a lot of analytical work. I have a few wishes to the authors, which, in my opinion, improve the article.

1) The abstract should be written more scientifically and introduce the reader in more detail to the course of the article.

2) The quality of Figure 2 needs to be improved. The legend should be more informative.

3) The conclusions should indicate the limitations of using nanocomposites

4) It would be good to reflect patented nanocomposites in the review so that the reader understands the development and applicability of the technology

Author Response

Dear reviewer:

We would like to thank the editor for giving us a chance to revise our paper, and also thank the reviewers for giving us constructive suggestions and comments which will help us to improve the quality of our submission. Here, we would like to submit our revised manuscript entitled “Recent advances in the synthesis, characterization, and application of nano-composites to the removal of endocrine-disrupting chemicals: a review”. Our response was attached below and the amendments have been highlighted on the revised manuscript.

Thank you for your re-evaluation of our revised submission. We hope that the revision will be acceptable for publication.

This is a very interesting article that will be of interest to a wide range of readers. The authors have done a lot of analytical work. I have a few wishes to the authors, which, in my opinion, improve the article.

  1. The abstract should be written more scientifically and introduce the reader in more detail to the course of the article.

Author reply: Thank you for your kind suggestion. The abstract of this manuscript was improved as suggested. The details could be found at lines 16 to 23.

  1. The quality of Figure 2 needs to be improved. The legend should be more informative.

Author reply: Thank the reviewer for giving us constructive suggestions. Figure 2 and the legend were revised.

  1. The conclusions should indicate the limitations of using nanocomposites

Author reply:The limitations and shortcomings of nanocomposites in water environment treatment were added, which can be seen at lines 543 to line 549 in the conclusions section.

  1. It would be good to reflect patented nanocomposites in the review so that the reader understands the development and applicability of the technology

Author reply: Thanks for your kind suggestion. The nanocomposite patents in recent years were supplied in the article, the details could be found in the line 435 to line 453.

Round 2

Reviewer 1 Report

The authors have added and corrected the manuscript from several aspects. I recommend for publication the corrected manuscript.

Author Response

Thank the reviewer for giving us constructive suggestion which will help us to improve the quality of our submission

Reviewer 2 Report

Unfortunately, the authors did not understand the point of my comments and did not address them in the right manner. 

Author Response

Thank the reviewer for giving us constructive suggestion which will help us to improve the quality of our submission. We tried our best to revise this manuscript according to your suggestions. We revise the title “Recent advances in the synthesis, characterization, and application of nano-composites to the removal of endocrine-disrupting chemicals: a review” to “Recent advances in the synthesis, characterization, and application of carbon nanomaterials to the removal of endocrine-disrupting chemicals: a review”. We focused the carbon nanomaterials for the removal of endocrine-disrupting chemicals. Most of the content of this revised manuscript has also been adjusted. The details of revision could be found in line 3-543.

Round 3

Reviewer 2 Report

The manuscript entitled “Recent advances in the synthesis, characterization, and application of carbon nanocomposites to the removal of endocrine-disrupting chemicals: a review” aims to review the nanocomposites that have been successfully used to remove EDCs from wastewater and also recent advances in nanocomposite synthesis and characterization.

The authors made some improvements, but some are still needed.

Here are my specific concerns:

The motivation for the selection of materials and EDCs is still not clear.

Adsorption mechanisms are still not needed here.

Application in wastewater – The authors write not only about carbon materials; this part was not narrowed down. It should be because, like this, it makes no sense.

The conclusion should be rewritten; it looks more like an abstract.  Follow some of the guidelines for scientific paper conclusion writing.

Figure 2 is not visible in full.

Author Response

We would like to thank the reviewer for giving us the third chance to revise our paper, and also thank the reviewer for giving us constructive suggestions which will help us to improve the quality of our revision. Here, we would like to submit our revised manuscript entitled “Recent advances in the synthesis, characterization, and application of carbon nanomaterials to the removal of endocrine-disrupting chemicals: a review”. Our response was attached below and the amendments have been highlighted on the revised manuscript.

The manuscript entitled “Recent advances in the synthesis, characterization, and application of carbon nanocomposites to the removal of endocrine-disrupting chemicals: a review” aims to review the nanocomposites that have been successfully used to remove EDCs from wastewater and also recent advances in nanocomposite synthesis and characterization.

The authors made some improvements, but some are still needed.

Here are my specific concerns:

1.The motivation for the selection of materials and EDCs is still not clear.

Author reply: Thank the reviewer for giving us constructive suggestions. The motivation for the selection of EDCs was explained at line 38-43, and line 45-72. The motivation for the selection of carbon nanomaterials was explained at line 121-152.

2.Adsorption mechanisms are still not needed here.

Author reply: The part of “adsorption mechanisms” have been all deleted.

3.Application in wastewater – The authors write not only about carbon materials; this part was not narrowed down. It should be because, like this, it makes no sense.

Author reply: Thank you for your kind suggestion. Only the carbon nanomaterials were described in part 5 “Application in wastewater”. The details could be found in 351-404.

  1. The conclusion should be rewritten; it looks more like an abstract. Follow some of the guidelines for scientific paper conclusion writing.

Author reply: We have serious about learning about scientific paper conclusion writing, the conclusion was revised and could be found in line 458-487.

5.Figure 2 is not visible in full.

Author reply: Thanks for your kind suggestion. Figure 2 was adjusted.

Round 4

Reviewer 2 Report

The authors addressed all my comments. I recommend this manuscipt for publication in present form.